# Microvascular flow alterations in critically ill COVID-19 patients: A prospective study

**Osama Abou-Arab**[1]*, **Christophe Beyls**[1], **Abdelilah Khalipha**[1], **Mathieu Guilbart**[1], **Pierre Huette**[1], **Stéphanie Malaquin**[1], **Benoit Lecat**[1], **Pierre-Yves Macq**[1], **Pierre Alexandre Roger**[1], **Guillaume Haye**[1], **Michaël Bernasinski**[1], **Patricia Besserve**[1], **Sandrine Soriot-Thomas**[2], **Vincent Jounieaux**[3], **Hervé Dupont**[1], **Yazine Mahjoub**[1]

1 Department of Anesthesiology and Critical Care Medicine, Amiens University Hospital, Amiens, France,
2 Department of Clinical Research, Amiens University Hospital, Amiens, France, 3 Department of Pneumology, Amiens University Hospital, Amiens, France

* osama.abouarab@gmail.com

**Data Availability Statement:** All relevant data are within the paper. Raw data are available after notification and authorization of the competent authorities. In France, all computer data (including

## Abstract

### Background

Data on microcirculatory pattern of COVID-19 critically ill patients are scarce. The objective was to compare sublingual microcirculation parameters of critically ill patients according to the severity of the disease.

### Methods

The study is a single-center prospective study with critically ill COVID-19 patients admitted in ICU. Sublingual microcirculation was assessed by IDF microscopy within 48 hours of ICU admission. Microcirculatory flow index (MFI), proportion of perfused vessel (PPV), total vessel density (TVD), De Backer score (DBS), perfused vessel density (PVD) and heterogeneity index (HI) were assessed. Patients were divided in 2 groups (severe and critical) according to the World health organization definition.

### Findings

From 19$^{th}$ of March to 7$^{th}$ of April 2020, 43 patients were included. Fourteen patients (33%) were in the severe group and twenty-nine patients (67%) in the critical group. Patients in the critical group were all mechanically ventilated. The critical group had significantly higher values of MFI, DBS and PVD in comparison to severe group (respectively, $PaCO_2$: 49 [44–45] vs 36 [33–37] mmHg; p<0,0001, MFI: 2.8 ± 0.2 vs 2.5 ± 0.3; p = 0.001, DBS: 12.7 ± 2.6 vs 10.8 ± 2.0 vessels mm$^{-2}$; p = 0.033, PVD: 12.5 ± 3.0 vs 10.1 ± 2.4 mm.mm$^{-2}$; p = 0.020). PPV, HI and TVD were similar between groups Correlation was found between microcirculatory parameters and $PaCO_2$ levels.

### Conclusion

Critical COVID-19 patients under mechanical ventilation seem to have higher red blood cell velocity than severe non-ventilated patients.

databases, in Cover Letter particular patient data) are protected by the National Commission on Informatics and Liberty (CNIL), the national data protection authority for France. CNIL is an independent French administrative regulatory body whose mission is to ensure that data privacy law is applied to the collection, storage, and use of personal data. As the database of this study was authorized by the CNIL, we cannot make available data without prior agreement of the CNIL. Requests may be sent to: picard.carl@chu-amiens. fr.

**Funding:** The study was not funded.

**Competing interests:** The authors have declared that no competing interests exist.

## Introduction

The most severe Coronavirus disease 19 (COVID-19) patients require critical care support in dedicated intensive care units (ICUs) to deliver oxygen supplemental and mechanical ventilation [1]. During COVID-19 outbreak, the Surviving Sepsis Campaign emitted recommendations on the management of critically ill patients: oxygen supplemental is required when oxygen saturation (SpO$_2$) is lower than 90% [2] and, if acute hypoxemic failure persists, tracheal intubation and mechanical ventilation are recommended. Nevertheless, clinical presentation of COVID-19 patients is uncommon and has been highlighted [3]. As the patient has a severe hypoxemia and compensated hyperventilation, no signs of fatigue or respiratory weakness are present probably due to the powerful hypocapnic ventilator inhibition [4, 5]. Timing for intubation and invasive ventilation in COVID-19 patients is a subject of debate [6]. In this new disease, microcirculatory impairment might explain disease progression and differences between invasively and non-invasively ventilated patients. [3]

Indeed, some reports have highlighted the increased lung perfusion in COVID-19 patients with probable intra-pulmonary shunt that is different from other causes of pneumonia [7, 8].

The aim of the study is to compare sublingual microcirculation of COVID-19 patients according the severity of the disease as defined by the World health organization [9].

## Materials and methods

### Ethics statement and population study

The study is an ancillary study of an ongoing single center observational study (CovidAmiens2020) conducted at Amiens Hospital University. According to French law on clinical research [10], the study was approved by our local institutional board (Comité de Protection des Personnes: chairman: Mr Salah Zerkly; registration identifier: PI2020_843_0026; approved on 15$^{th}$ of Mars 2020) and was registered on ClinicalTrial.gov on 21$^{st}$ of April 2020 (identifier: NCT04354558). The Agence nationale de security et du médicament notification was not required about our study. This study was conducted in accordance with the amended Declaration of Helsinski. Regarding French laws, our study was classified as a category 3 [11]. Thus, written consent was waved. The patient was orally informed in his right to oppose to the use of his data after delivering a written information on the study. A copy of the written information with the patient name was joined to the medical record.

All patients over 18 with COVID-19 disease confirmed by rt-PCR on a nasopharyngeal swab and admitted to our intensive care unit (ICU) for respiratory support were prospectively included.

### Definition of severe and critical group

The severity was defined according the Worldwide Health Organization for COVID-19 case definition [9]. The patient was defined as severe (Severe group) when having a respiratory rate $\geq$ 30 breaths/min or oxygen saturation $\leq$ 90% on room air or signs of severe distress syndrome. The patient was defined as critical (Critical group) when having respiratory failure and requiring mechanical ventilation or shock or organ failure that requires ICU care.

### Microcirculation assessment

Microcirculation was assessed 48 hours within ICU admission. An incident dark field imaging device (CytoCam$^\circledR$, Braedius Medical, Huizen, the Netherlands) was used to assess sublingual microcirculation. We measured the microvascular flow index (MFI), the total vessels density (TVD), the proportion of perfused vessels (PPV), the De Backer score (DBS) and the

heterogeneity index (HI). As recommended by the European society task force of intensive care medicine in their second consensus, five video sequences of twenty seconds each, were recorded from 4 different sublingual areas for each patients [12]. The microcirculation image quality was assessed using the scoring system proposed by Massey et al [13].

Off record digital videos were analyzed in a semi-automated manner with a dedicated software (Automated Vascular Analysis 3.2, Microvision Medical, Amsterdam, the Netherlands) by an independent investigator blind to the patient characteristics. Only vessels with a diameter below 20 μm were assessed.

### Data collection

For each patient, we collected demographic and biological data at ICU admission.

At the time of microcirculation assessment, blood gases, critical care severity scores, respiratory ventilation type, echocardiographic data (visual left ventricular ejection fraction and cardiac output), hemodynamics data (mean arterial pressure, heart rate, vasopressor use, capillary refill time and mottling score) were collected. All COVID-19 patients were under unfractioned heparin to achieve anti Xa activity between 0.5 and 0.8 UI/ml"

"Capillary refill time was performed as follows: a firm pressure was applied to the surface of the index finger distal phalanx. The pressure was increased until the skin became blank and maintained for 10s. The pressure was then released. The time for return to pre-existent skin color was measured using a chronometer [14].

Cardiac output was measured by Doppler echocardiography by the same operator (CB)."

### Statistical analysis

No sample size was assessed for the study given the absence of data on Cytocam parameters in Covid-19 at the time of inclusion. We included the maximum of patients we could during the first wave of COVID-19 outbreak.

Data were presented as mean (standard deviation), median [interquartile range] or as numbers (percentage). Severe and critical groups were compared using a Mann-Whitney test, a chi-2 test or a Fischer exact test, as appropriate. Effect size on microcirculatory parameters was performed using Cohen D coefficient. Cohen D coefficient less than 0.2 was considered as small (less than 0.2), moderate (near 0.5), or large (more than 0.8) [14]. Correlations between microcirculatory parameters and blood gas were performed using a Spearman correlation test. A p value<0.05 for a statistical test was considered as significant. Statistical analysis was performed using SPSS software version 24 (IBM Corp, Armonk, NY).

## Results

### Demographic data (Table 1)

From 19th of March to 7th of April 2020, 43 patients were included: 14 in the severe group (33%) and 29 in the critical group (67%).

Age, body mass index and comorbidities were similar between groups. Duration from the symptoms onset to ICU admission was similar between groups (8 [4–10] vs 7 [5–10] days; p = 0.894 for critical and severe groups, respectively). SOFA and SAPS II scores at inclusion were significantly higher in critical group when compared to severe group.

### Biological investigations at ICU admission (Table 1)

The critical group showed a higher C reactive protein level (217 [156–313] vs 152 [113–189]; p = 0.049), a lower lymphocyte count (700 [600–1050] vs 1000 [800–1300] mm$^{-3}$; p = 0.015)

**Table 1. Demographics between severe and critical patients.**

| Variables | Severe group (n = 14) | Critical group (n = 29) | P value |
|---|---|---|---|
| Age; *years* | 62 [50–68] | 63 [57–68] | 0.343 |
| Male gender; *n (%)* | 12 (86) | 26 (90) | 1.000 |
| BMI; *kg m$^{-2}$* | 27.8 [24.3–34.7] | 30.1 [29.2–33.2] | 0.067 |
| Medical history; *n (%)* | | | |
| Hypertension | 6 (42) | 17 (59) | 0.442 |
| Diabetes | 4 (29) | 4 (14) | 0.404 |
| Dyslipidemia | 4 (29) | 5 (17) | 0.442 |
| Severe obesity | 1 (7) | 5 (17) | 0.645 |
| COPD/Asthma | 0 | 1 (3) | 1.000 |
| Days from symptom onset to ICU admission; *days* | 8 [5–10] | 5 [2–8] | **0.003** |
| Days from ward to ICU admission; *days* | 4 [2–5] | 2 [1–2] | **0.02** |
| Time from hospital admission to intubation; *hours* | - | 2 [1–2] | - |
| Biological investigations | | | |
| WBC; *mm$^{-3}$* | 6400 [5800–8400] | 8300 [7250–1050] | 0.053 |
| Lymphocyte count; *mm$^{-3}$* | 1000 [800–1300] | 700 [600–1050] | **0.015** |
| Hemoglobin; *g l$^{-1}$* | 11.9 [11.0–12.7] | 11.3 [10.2–12.5] | 0.211 |
| C reactive protein; *mg l$^{-1}$* | 152 [113–189] | 217 [156–313] | **0.049** |
| Procalcitonin; *ng ml$^{-1}$* | 0.39 [0.08–0.59] | 1.47 [0.42–2.66] | **0.012** |
| Platelet count; *$10^3$ mm$^{-3}$* | 256 [235–313] | 206 [150–321] | 0.500 |
| D-dimer; *μg l$^{-1}$* | 2.5 [1.5–5.06] | 4.7 [1.0–7.9] | 0.901 |
| Fibrinogen; *g l$^{-1}$* | 6.2 [5.2–6.6] | 6.9 [5.2–8.0] | 0.124 |
| Soluble fibrin complex; *μg ml$^{-1}$* | 4.0 [3.6–4.3] | 4.3 [3.7–6.0] | 0.554 |
| *Respiratory support; n (%)* | | | |
| Oxygen mask | 11 (79) | - | |
| HFNC | 3 (21) | - | |
| Mechanical ventilation | - | 29 (100) | |
| *CT scan at hospital admission* | | | |
| Ground glass opacities; *n (%)* | 12 (86) | 26 (90) | 0.628 |
| Crazy paving; *n (%)* | 4 (29) | 8 (27) | 0.836 |
| Consolidation; *n (%)* | 7 (50) | 15 (52) | 0.882 |
| Pulmonary embolism; *n (%)* | 0 (0) | 1 (3) | 0.934 |
| SOFA score at inclusion | 3 [1–4] | 10 [7–13] | <0.0001 |
| SAPS II score at inclusion | 30 [23–32] | 65 [55–71] | <0.0001 |

**COPD**: chronic obstructive pulmonary disease; **HFNC**: high flow nasal oxygen cannula, **SAPS II**: simplified acute physiology score II, **SOFA**: sequential organ failure assessment. Data were expressed as median [interquartile range] or as number (percentage). Data were compared using Mann-Whitney, Chi-2 or a Fischer exact test.

and a higher procalcitonin level (1.47 [0.42–2.66] vs 0.39 [0.08–0.59] ng ml$^{-1}$) when compared to the severe group. White blood and platelet counts were similar in both groups. Both groups presented a similar coagulopathy with comparable high D-dimer and soluble complex levels.

## Hemodynamics parameters, blood gas and ventilation at inclusion (Table 2)

MAP, HR, cardiac output and SpO$_2$ were similar between groups. No significant difference in capillary refill time or mottling score was observed between groups.

PaO$_2$ levels were similar between groups. The severe group presented a respiratory alkalosis compared to the critical group with a significant higher pH (respectively, 7.46 [7.45–7.48] vs

**Table 2. Respiratory, biological and microcirculatory data between severe and critical patients.**

| Variables | Severe group (n = 14) | Critical group (n = 29) | P value |
|---|---|---|---|
| HR; *bpm* | 96 [76–114] | 87 [68–94] | 0.468 |
| MAP; *mm Hg* | 95 [76–98] | 81 [73–92] | 0.117 |
| SpO$_2$; *%* | 98 [96–100] | 95 [94–96] | 0.239 |
| Cardiac output; *l min$^{-1}$* | 5.6 [4.0–6.6] | 5.0 [4.0–6.6] | 0.642 |
| LVEF; *%* | 60 [59–70] | 55 [50–60] | 0.085 |
| Norepinephrine use; *n (%)* | 0 (0) | 13 (45) | **0.003** |
| Median dose; *µg.kg.min$^{-1}$* | - | 0.02 | NA |
| CRT; *sec* | 2 [1–3] | 3 [2–4] | 0.06 |
| Mottling score | | | |
| 0 | 13 (93) | 17 (59) | |
| 1 | 0 (0) | 7 (24) | |
| 2 | 1 (7) | 3 (10) | 0.110 |
| 3 | 0 (0) | 0 (0) | |
| >3 | 0 (0) | 2 (7) | |
| *Blood gases analysis* | | | |
| pH | 7.46 [7.45–7.48] | 7.36 [7.3–7.38] | **<0.0001** |
| PaCO$_2$; *mmHg* | 36 [33–37] | 49 [44–55] | **<0.0001** |
| PaO$_2$; *mmHg* | 79 [71–110] | 84 [74–100] | 0.746 |
| Lactate; *mmol l$^{-1}$* | 1.5 [1.1–1.7] | 2.0 [1.7–2.5] | **0.001** |
| *Ventilation settings* | | | |
| Compliance; *ml H$_2$O$^{-1}$* | - | 41 [32–44] | - |
| PEEP; *cmH$_2$O* | - | 14 [12–15] | - |
| Tidal volume; *ml kg$^{-1}$* | - | 5.5 [4.6–6.3] | - |
| PaO$_2$/FiO$_2$ | - | 131 [91–145] | - |
| Plateau pressure; *cmH$_2$O* | - | 25 [23–26] | - |
| Time from ICU admission to | | | |
| inclusion; *hours* | 20 [5–32] | 21 [6–30] | 0.682 |
| HFNO before inclusion; *hours* | 20 [5–32] | - | - |
| *Microcirculatory parameters* | | | |
| MFI | 2.5 ± 0.3 | 2.8 ± 0.2 | **0.001** |
| PVD; *mm.mm$^{-2}$* | 10.1 ± 2.4 | 12.5 ± 3.0 | **0.020** |
| PPV; *%* | 98 ± 3 | 94 ± 9 | 0.090 |
| TVD; *mm.mm$^{-2}$* | 13.8 ± 3.1 | 15.5 ± 3.3 | 0.224 |
| DBS; *n vessels.mm$^{-2}$* | 10.8 ± 2.0 | 12.7 ± 2.6 | **0.033** |
| Heterogeneity index | 0.05 ± 0.06 | 0.1 ± 0.1 | 0.189 |
| Went for mechanical ventilation within the 7 next days; *n (%)* | 7 (50) | - | - |
| Duration of intubation; *days* | 20 [15–29] | 22 [16–32] | 0.104 |
| ICU mortality; *n (%)* | 1 (8) | 7 (24) | **<0.0001** |
| Discharge from ICU; *n (%)* | 13 (93) | 21 (72) | **<0.0001** |
| Discharge from hospital; *(n%)* | 11 (79) | 19 (66) | **<0.0001** |

**HR:** heart rate; **MAP:** mean arterial pressure; **LVEF:** left ventricular ejection fraction; **CRT:** capillary refill time; **PEEP:** positive end expiratory pressure; **ICU:** intensive care unit; **MFI:** microvascular flow index; **PVD:** perfused vessel density; **TVD:** total vessel density; **PPV:** proportion of perfused vessel; **DBS**: De Backer Score; **NA:** non-applicable. HFNO: High flow nasal oxygen. Data were expressed as median [interquartile range] or as number (percentage). Data were compared using Mann-Whitney, Chi-2 or a Fischer exact test.

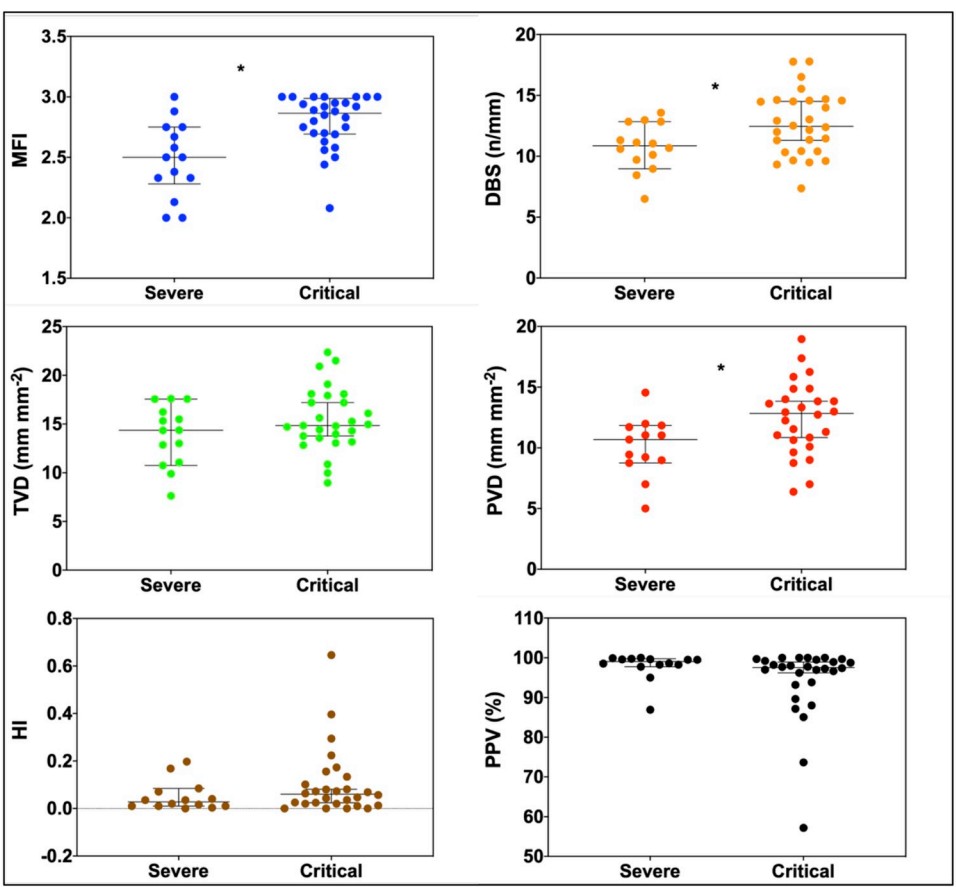

**Fig 1. Comparisons of microcirculatory flow index (MFI), total vessel density (TVD), perfused vessel density (PVD), proportion of perfused vessel (PPV), De Backer score (DBS) and heterogeneity index (HI) in severe group and critical group.** *: P value <0.05 between groups comparisons using Mann-Whitney U test.

7.36 [7.3–7.38]; p<0.0001) and a significant lower $PaCO_2$ (respectively, 36 [33–37] vs 49 [44–55] mmHg; <0.0001).

## Microcirculation assessment (Table 2 and Fig 1)

In the critical group, MFI, PVD and DBS were significantly higher in comparison to the severe group (respectively, for critical and severe groups, MFI: 2.8 ± 0.2 vs 2.5 ± 0.3, p = 0.001, D = 1.16; PVD: 12.5 ± 3.0 vs 10.1 ± 2.4 mm.mm$^{-2}$, p = 0.02, D = 0.14 and DBS: 12.7 ± 2.6 vs 10.8 ± 2.0 vessels.mm$^{-2}$, p = 0.033, D = 0.08).

PPV (p = 0.09, D = 0.05), TVD (p = 0.224, D = 0.55) and HI (p = .189, D = 1.15) were similar for both groups.

## Correlation between $PaO_2$, $PaCO_2$ and microcirculation (Fig 2)

$PaCO_2$ appeared significantly correlated to MFI (Rho = 0.428; p = 0.005), PVD (Rho = 0.363; p = 0.023) and DBS (Rho = 0.276; p = 0.048) but not with PPV (Rho = 0.010; p = 0.948) TVD (Rho = 0.194; p = 0.224) and HI (Rho = -0.045; p = 0.777).

No significant correlations were found between $PaO_2$ and MFI (Rho = 0.21; p = 0.893), PVD (Rho = -0.76; p = 0.644), DBS (Rho = -0.99; p = 0.534), PPV (Rho = -0.263; p = 0.092), TVD (Rho = -0.91; p = 0.573) and HI (Rho = 0.198; p = 0.209).

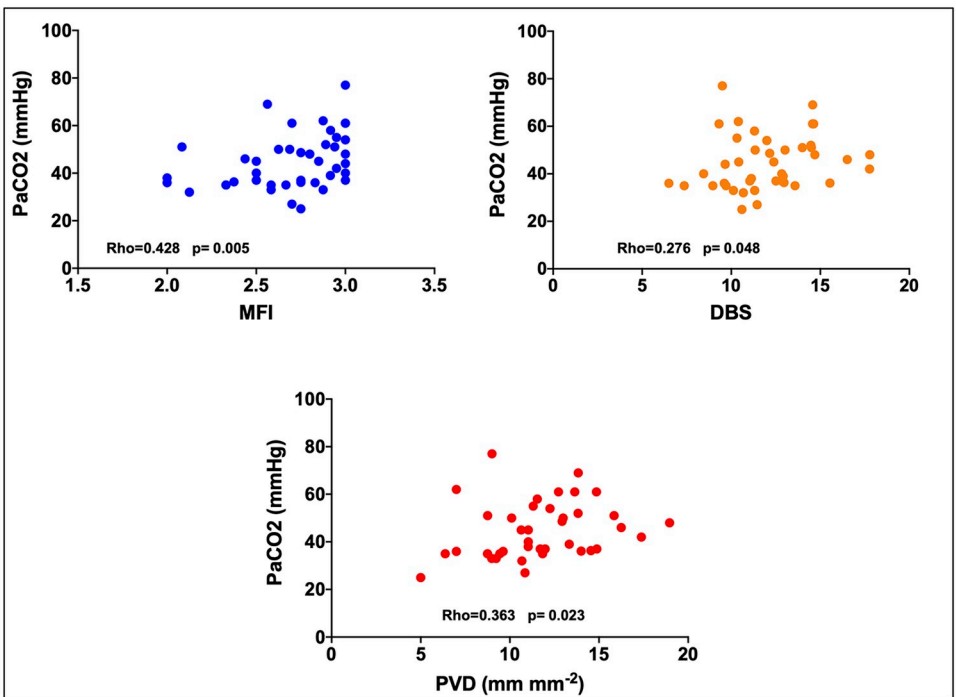

**Fig 2. PaCO$_2$ correlation with the microcirculatory flow index (MFI), the perfused vessel density (PVD) and the De Backer score (DBS).**

## Discussion

Our findings suggest that the critical group had higher red blood cell velocity (higher MFI values) and a better vessel density (higher DBS and PVD). Capillary diffuse properties seem to be preserved given the values of PPV, TVD and HI.

Theses phenomenon do not appear related to PaO$_2$ which is similar in both groups. PaCO$_2$ levels seem to be correlated with microcirculation parameters.

We could have expected more altered microcirculatory parameters in the critical group than in the severe group with lower red blood cell velocities and a lower vessel density regarding the higher pro inflammatory state (higher C reactive protein and procalcitonin levels) and the higher severity scores (SAPS II and SOFA). Indeed, we learned from severe sepsis and septic shock that microcirculation in critically ill patients is usually impaired with an increase in heterogeneity, a decrease in vessel density and perfusions indices [15, 16]. It has been shown that vessel density and DBS are correlated with mortality in a large cohort of sepsis [17]. However, our critical group did not present the typical characteristics of septic shock (hyperkinetic cardiac state, high lactate level and high dose of norepinephrine. In our series, COVID-19 patients presented an isolated respiratory failure, sometimes associated with an acute kidney injury but with limited need for vasopressors and no clinical signs of hypo perfusion. These data are in accordance with previous reports on critically ill COVID-19 patients showing a low prevalence of septic shock or multi organ failure [18].

In a previous microcirculatory study on 9 COVID-19 patients under ECMO therapy, Carsetti et al. found that TVD and PVD were inversely correlated with D-dimer [19]. The same team found similar results on 12 COVID-19 patients under mechanical ventilation [20]. Values of microcirculatory parameters are similar to ours. However, we did not found any difference in D-dimer levels between our 2 groups.

Our results differ from other studies on severe virus's infections. In a report on mechanically ventilated patients with Influenza infection, Salgado et al found a compromised microcirculation with a median MFI of 1.9 [21].

Hence, to explain our results, we focused on differences between respiratory parameters. Finding that MFI, PVD and DBS, the three most significant parameters, were correlated to $PaCO_2$ levels (and hence to the type of ventilation) we hypothesized that hypocapnia might influence systemic microcirculation. In an experimental study on rabbits, Komori et al. have showed that systemic microcirculatory flow decreases with hypocapnia and is restored with the increase of $PaCO_2$ (up to a certain threshold). Same findings are reported in a dog model for which hypercapnia improves vascular capacitance [22]. Moreover, hypocapnia is known to increase the hemoglobin affinity for oxygen, moving the hemoglobin-oxygen dissociation curve to the left and thus decreasing oxygen supply to organs.

In our series, there is no differences between groups concerning $PaO_2$ and we did not find any correlation between microcirculatory parameters and $PaO_2$. Several studies have shown that hypoxia enhances microcirculatory parameters trough a physiological adaptation to match oxygen delivery to demand [23]. So, microvascular blood flow increases to adapt to hypoxemic state.

Clinical studies on the effect of mechanical ventilation on microcirculation are scarce. In a recent physiologic paper, Ospina-Tascon et al. showed that sublingual microcirculation heterogeneity may decrease by reducing dead space ventilation in patient with ARDS [24]. The mechanism remains unclear but in a context of sepsis, heterogeneity in perfusion could alter both lung and systemic perfusions which seem intricately linked [25]. In our study, we did not find any difference between groups in heterogeneity index. Our hypothesis concerning the effect of $PaCO_2$ on microcirculation needs further confirmation in clinical studies as our correlations (given Spearman Rho values) were moderate or weak and cofounders may not be excluded.

Another explanation is the higher dose of norepinephrine in the critical group with a potential capillary recruitment (Table 2). However, the effect of norepinephrine on microcirculation is still unclear. Jhanji *et a*. showed an increase in oxygen delivery with norepinephrine in patients with septic shock without improving MFI [26]. In a prospective randomized trial, Dubin *et al.* concluded to the lack of effect of norepinephrine on microcirculation [27]. However, once again, our pattern of microcirculation differed from septic shock and an absence of norepinephrine effect cannot be excluded even if parameters as cardiac index and mean arterial pressure were similar between groups. Moreover, the higher DBS, MFI and PVD values in most severe patients could reflect a state of loss in vascular tone with a more apparent hyperemic state. Pulmonary imaging and postmortem studies of COVID-19 patients have highlighted the hypothesis of an increased pulmonary blood flow with intrapulmonary vascular shunting responsible for hypoxemia [5, 7, 8]. Our data may reflect this hyperemic state.

Our study has several limitations. First, we recorded only one time-point measurement and, to confirm our results, different microcirculatory assessments, especially before and after tracheal intubation, would have been of great interest. A new tool called MicroTools software allowing an automated video analysis have been recently validated in a large international retrospective database [28]. This kind of automated analysis allowing resuscitation therapies targeted on microcirculatory parameters should be applied in further investigations. Second, we did not perform a sample size calculation. However, we applied the Cohen D test to assess the size effect of each parameter. Except for MFI, all parameters had a D value under 0.2 suggesting the absence of size effect making p value interpretation more reliable. Last, as patients in the sever group were spontaneously breathing, difficulty in sublingual assessment in awake patients may have influenced the results. However, the Massey score for image quality was

comparable in the 2 groups. Finally, we did not evaluate central venous oxygen saturation which could be valuable to compare balance between oxygen consumption and oxygen delivery or you can say that we did not record central venous oxygen derived variables to compare oxygen consumption.

## Conclusion

In this limited series of critically ill COVID-19 patients, microcirculation did not show heterogeneity as observed in septic shock. Moreover, patients under mechanical ventilation with higher severity scores but with higher $PaCO_2$ levels, had higher red blood cell velocities as assessed by MFI. Further controlled clinical investigations are required to assess the effect of hypocapnia on microcirculation in this setting.

## Acknowledgments

We thank Amiens university hospital staff for their support in the study process and their commitment in data collection during the pandemic.

## Author Contributions

**Conceptualization:** Osama Abou-Arab, Vincent Jounieaux, Yazine Mahjoub.

**Data curation:** Christophe Beyls, Abdelilah Khalipha, Mathieu Guilbart, Pierre Huette, Stéphanie Malaquin, Benoit Lecat, Pierre-Yves Macq, Pierre Alexandre Roger, Guillaume Haye, Michaël Bernasinski, Patricia Besserve, Hervé Dupont, Yazine Mahjoub.

**Resources:** Sandrine Soriot-Thomas.

**Writing – original draft:** Osama Abou-Arab, Vincent Jounieaux, Yazine Mahjoub.

**Writing – review & editing:** Osama Abou-Arab, Vincent Jounieaux, Hervé Dupont, Yazine Mahjoub.

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
