## [Decision Letter · Decision Letter 0]

24 Nov 2020

PONE-D-20-32823

Systemic microcirculatory patterns of critically ill Covid-19 patients: a prospective study

PLOS ONE

Dear Dr. abou arab,

Thank you for submitting your manuscript to PLOS ONE. After careful consideration, we feel that it has merit but does not fully meet PLOS ONE’s publication criteria as it currently stands. Therefore, we invite you to submit a revised version of the manuscript that addresses the points raised during the review process.

We look forward to receiving your revised manuscript.

Kind regards,

Yu-Chang Yeh, M.D., Ph.D.

Academic Editor

PLOS ONE

Additional Editor Comments:

Please refer to the reviewers' comments and revise your manuscript as appropriate. Keep safe and all the best.

Journal Requirements:

2.Thank you for including your ethics statement:

"According to French law on clinical research [11], the study was declared and registered at the Commission Numérique Informatiques et Libertés (registration identifier:

I2020_843_0026) and was registered on ClinicalTrial.gov on 21st of April 2020 (identifier: NCT04354558). This study was conducted in accordance with the amended Declaration of Helsinski."   

3. In the Methods, please state:

- Why written consent could not be obtained

- Whether the Institutional Review Board (IRB) approved use of oral consent

- How oral consent was documented

For more information, please see our guidelines for human subjects research: https://journals.plos.org/plosone/s/submission-guidelines#loc-human-subjects-research

4. Please provide a sample size and power calculation in the Methods, or discuss the reasons for not performing one before study initiation.

"NO- The funders had no role in study design, data collection and analysis, decision to publish, or preparation of the manuscript."

6. We note that you have indicated that data from this study are available upon request. PLOS only allows data to be available upon request if there are legal or ethical restrictions on sharing data publicly. For information on unacceptable data access restrictions, please see http://journals.plos.org/plosone/s/data-availability#loc-unacceptable-data-access-restrictions.

7. Please ensure that you refer to Figure 1 and 2 in your text as, if accepted, production will need this reference to link the reader to the figure.

Reviewers' comments:

Reviewer's Responses to Questions

**Comments to the Author**

1. Is the manuscript technically sound, and do the data support the conclusions?

Reviewer #1: Yes

Reviewer #2: Yes

2. Has the statistical analysis been performed appropriately and rigorously? 

Reviewer #1: Yes

Reviewer #2: Yes

3. Have the authors made all data underlying the findings in their manuscript fully available?

Reviewer #1: Yes

Reviewer #2: Yes

4. Is the manuscript presented in an intelligible fashion and written in standard English?

Reviewer #1: Yes

Reviewer #2: Yes

5. Review Comments to the Author

Reviewer #1: The authors showed interesting microcirculatory patterns in COVID-19 patients. I have some comments and questions :

1) The authors mentioned that sublingual microcirculation was assessed within 48 hours after ICU admission. What about the median inclusion time ? Did sublingual microcirculatory assessment was faster in critical group than severe group? (p.7)

2) For data collection for systemic hemodynamic variables, Was cardiac output measured by echocardiography in all patients ? if yes , was echocardiography performed by 1 persons due to operator dependent? For capillary refill time, could you mention how did you performed capillary refill time by eye ball or using chronometer ?

3) For demographic data, were SOFA and SAPS II scores calculated at ICU admission or time of inclusion ? It would be better if you mention severity scores at inclusion time. Was PaO2/FiO2 ratio in severe group different from patients in critical group. The authors mentions number of patients who required vasopressor, in these patients, were they shock ? Could you possibly mention numbers of patients with shock and without shock and did microcirculatory profiles differ between patients with and without shock.

4) Was the microcirculatory profiles related to lung recovery in critical group ?

5) Did microcirculatory alter in patients with COVID19 compare with healthy volunteers?

Reviewer #2: I would like to congratulate the authors for their courage and enthusiasm in performing such time consuming and labor intensive study protocol in the ICU.

I have just minor concerns:

1. The title could be more focussed beginning by deleting the word "systemic". Please rewrite the title specifically by adding more microcirculatory items like capillary, microvascular flow, alterations etc...

2. Please write COVID-19 with capital letters everywhere.

3. Abstract: Conclusion should be specified by replacing "better microcirculatory pattern" to more focussed description. What is better???? What is still wrong in the microcirculation???

4. Introduction: "Microcirculation pattern of Covid-19 might differ from other sepsis causes" could implicate that COVID-19 is (pre)sepsis. Please rewrite this sentence that doesn't confuse sepsis with this new disease.

5. Methods: MFI=microvascular flow index

6. Results: Please refer to figure 1 and 2 for the responsible data.

7. Discussion: First sentence is very superficial. Higher microcirculation parameters does mean nothing. Please redifine this e.g. hyperdynamic at least?? High density vasculary flows? high red blood cell velocity?

8. From limitations you could delete the third limitation hence there are thousands of studies which support the idea of sublingual microcirculation as the best paramter for follow-up of the end-organ perfusion ("systemic

(micro)circulation").

9. Finaly add in your discussion the more advanced novel automated microcirculation analysis which could be used for more detailed and accurate microcirculatory analysis published by Hilty et al. Crit Care Med 2020. Novel studies with novel and efficient microcirculation analysis programs could make the clinicians ICU life scientifically comfortable.

Tables: Please add more clinical data into the tables

- CT lungs for detecting pulmonary embolism; how many performed how many positive in each group?

- Day of illness?

- Day of intubation?

- Day of optiflow during the microcirculation measurement?

- What was the anticoagulation strategy? Profylaxes, therapeutic? which agent? UFH? LMWH? What was the follow-up/monitoring strategy for coagulation?

6. PLOS authors have the option to publish the peer review history of their article (what does this mean?). If published, this will include your full peer review and any attached files.

Reviewer #1: No

Reviewer #2: No

---

## [Author Response · Author response to Decision Letter 0]

21 Dec 2020

Reviewer #1: The authors showed interesting microcirculatory patterns in COVID-19 patients. I have some comments and questions:

1) The authors mentioned that sublingual microcirculation was assessed within 48 hours after ICU admission. What about the median inclusion time?

Answer: We thank the reviewer for his comment. We checked for ICU exact admission time on our electronic health record software and match it with the Cytocam software record for each patient. 

The median time inclusion was similar between group, around 20 hours. We added it in table 2 as follows:

Time from ICU admission to inclusion; hours 20 [5-32] 21 [6-30] 0.682

Comment: Did sublingual microcirculatory assessment was faster in critical group than severe group? (p.7)

Answer: Regarding the assessment, we proceeded with the same manner whatever the patient severity. We took the necessary time to record 5 videos of good quality as required by the semi-automatic software analysis. In cases of bad quality of videos, we tried as many attempts as necessary to obtain 5 videos of good quality. We did not specifically record the assessment duration. However, it seems to the operator that the duration of the assessment was faster in sedated patients compared to awake patients.

2) Comment For data collection for systemic hemodynamic variables, Was cardiac output measured by echocardiography in all patients ? if yes , was echocardiography performed by 1 persons due to operator dependent? For capillary refill time, could you mention how did you performed capillary refill time by eye ball or using chronometer ?

Answer: We thank the reviewer for his comment.

Cardiac output was measured by Doppler echocardiography. All measurements for all patients were performed by the same operator (CB). We added a sentence in the method section: “Cardiac output was measured by Doppler echocardiography by the same operator(CB).”

Regarding capillary refill time, measurements were standardized using a chronometer. We added details on the standardization of measurement as follows:

“Capillary refill time was performed as follows: a firm pressure was applied to the surface of the index finger distal phalanx. The pressure was increased until the skin became blank and maintained for 10s. The pressure was then released. The time for return to pre-existent skin color was measured using a chronometer [14].”

3) For demographic data, were SOFA and SAPS II scores calculated at ICU admission or time of inclusion ? It would be better if you mention severity scores at inclusion time.

Answer: We agree with reviewer. SAPS II and SOFA were calculated at inclusion time (time of microcirculation assessment) as mentioned in data collection section. We added “at inclusion” in the results section and in Table 1.

Comment: Was PaO2/FiO2 ratio in severe group different from patients in critical group. 

Answer: We thank the reviewer for his relevant comment. However, in the severe group patients had a facial mask or a nasal canula which makes the exact estimation of FiO2 difficult. That is why we apologize for not being able to report this value for the severe group.

Comment: The authors mentions number of patients who required vasopressor, in these patients, were they shock? Could you possibly mention numbers of patients with shock and without shock and did microcirculatory profiles differ between patients with and without shock.

Answer: We thank the reviewer for his relevant comment. No patient had a shock state at inclusion. As reported in table 2, the dose of norepinephrine was quite low (0.02 μg.kg.min-1). Macro hemodynamics parameters were similar regarding cardiac output and heart rate with no septic shock pattern. Vasopressors use was probably related to sedation drugs (propofol) used to ensure mechanical ventilation. 

4) Was the microcirculatory profiles related to lung recovery in critical group?

Answer: We thank the reviewer for his relevant comment. Unfortunately, only one time point measurement was performed. We agree that several a time point monitoring of microcirculation during the stay would had been very interesting. However, microcirculatory record was performed shortly after initiation of mechanical ventilation( i.e: worsening of lung function). 

5) Did microcirculatory alter in patients with COVID19 compare with healthy volunteers?

Answer: We thank the reviewer for his relevant comment.

The guidelines on sublingual assessment (doi: 10.1007/s00134-018-5070-7) described the convective property according to the red blood cell velocities (explored by the MFI and the PVD), the diffuse capacity (explored by the TVD) and the heterogeneity in perfusion (HI)

No reference values were admitted but healthy subjects values are available from previous studies (doi: 10.1097/CCM.0b013e31823dae59).

In a healthy subject, a normal MFI is expected to be over 2.6 which was the case for our critical group but not in the severe group. 

PVD in healthy subject was reported to be about 15 mm/mm2. PVD was less decreased in the severe group.

To summarize, the critical group tend to have higher red blood velocities when comparing the severe group (PVD and MFI higher)

In a healthy subject, TVD is around 15 mm/mm2 and HI around 0.03. In our population, these parameters were close to healthy subjects’ values. Hence, in our cohort COVID-19 patients did not seem to have convective impairment.

Reviewer #2: I would like to congratulate the authors for their courage and enthusiasm in performing such time consuming and labor intensive study protocol in the ICU.

Answer: we are very grateful to the reviewer for his consideration and encouragements.

I have just minor concerns:

1. The title could be more focussed beginning by deleting the word "systemic". Please rewrite the title specifically by adding more microcirculatory items like capillary, microvascular flow, alterations etc...

Answer: We thank the reviewer for his helpful comment. We changed the title as follows: “Microvascular flow alterations in critically ill COVID-19 patients: a prospective study.”

2. Please write COVID-19 with capital letters everywhere.

Answer: We made changes in the manuscript.

3. Abstract: Conclusion should be specified by replacing "better microcirculatory pattern" to more focussed description. What is better???? What is still wrong in the microcirculation???

Answer: We thank the reviewer for his comment and made changes to better describe the profiles. We made changes as follows:

Critical COVID-19 patients under mechanical ventilation seem to have higher red blood cell velocity than severe non-ventilated patients.

4. Introduction: "Microcirculation pattern of Covid-19 might differ from other sepsis causes" could implicate that COVID-19 is (pre)sepsis. Please rewrite this sentence that doesn't confuse sepsis with this new disease.

“Timing for intubation and invasive ventilation in COVID-19 patients is a subject of debate [6]. In this new disease, microcirculatory impairment might explain disease progression and differences between invasively and non-invasively ventilated patients. [3] 

5. Methods: MFI=microvascular flow index

Answer: We apologize for this mistake and made changes

6. Results: Please refer to figure 1 and 2 for the responsible data.

Answer: We made changes and referred to figure 1 and 2 when required in the result section.

7. Discussion: First sentence is very superficial. Higher microcirculation parameters does mean nothing. Please redifine this e.g. hyperdynamic at least?? High density vasculary flows? high red blood cell velocity?

Answer: We thank the reviewer for his relevant comments and hope we better describe the profiles.

Here we made the following changes.

“Our findings suggest that the critical group had higher red blood cell velocity (higher MFI values) and a better vessel density (higher DBS and PVD). Capillary diffuse properties seem to be preserved given the values of PPV, TVD and HI.”

8. From limitations you could delete the third limitation hence there are thousands of studies which support the idea of sublingual microcirculation as the best paramter for follow-up of the end-organ perfusion ("systemic (micro)circulation").

Answer: We made changes and removed the sentence.

9. Finaly add in your discussion the more advanced novel automated microcirculation analysis which could be used for more detailed and accurate microcirculatory analysis published by Hilty et al. Crit Care Med 2020. Novel studies with novel and efficient microcirculation analysis programs could make the clinicians ICU life scientifically comfortable.

Answer: We thank the reviewer for his comment and added a sentence explaining the MicroTools.

“A new tool called MicroTools software allowing an automated video analysis have been recently validated in a large international retrospective database [29]. This kind of automated analysis allowing resuscitation therapies targeted on microcirculatory parameters should be applied in further investigations.” 

Comments 

Tables: Please add more clinical data into the tables

- CT lungs for detecting pulmonary embolism; how many performed how many positive in each group?

- Day of illness?

- Day of intubation?

- Day of optiflow during the microcirculation measurement?

Answer: We thank the reviewer for his comment and completed Table 1 and Table 2 as follows.

- What was the anticoagulation strategy? Profylaxes, therapeutic? which agent? UFH? LMWH? What was the follow-up/monitoring strategy for coagulation?

Answer: during the first wave of COVID-19 pandemic, our local institutional protocol included for all COVID-19 patients admitted to ICU, a therapeutic anticoagulation strategy with unfractioned heparin to achieve anti Xa activity between 0.5 and 0.8 UI/ml.

 UFH was started at ICU admission. Anti Xa activity was checked daily and 4 hours after each dose change. We added a sentence in the method section “All COVID-19 patients were under unfractionned heparin to achieve anti Xa activity between 0.5 and 0.8 UI/ml”

---

## [Decision Letter · Decision Letter 1]

15 Jan 2021

PONE-D-20-32823R1

Microvascular flow alterations  in critically ill COVID-19 patients: a prospective study

PLOS ONE

Dear Dr. abou arab,

Thank you for submitting your manuscript to PLOS ONE. Therefore, we invite you to submit a revised version of the manuscript that addresses the points raised during the review process.

We look forward to receiving your revised manuscript.

Kind regards,

Yu-Chang Yeh, M.D., Ph.D.

Academic Editor

PLOS ONE

Additional Editor Comments:

Thank you for carefully revising your manuscript. We have received the reply from all reviewers.

Please revise the sentence mentioned by the reviewer.

Please consider drawing a dot-boxplot to refine Figure 1.

Reviewers' comments:

Reviewer's Responses to Questions

**Comments to the Author**

1. If the authors have adequately addressed your comments raised in a previous round of review and you feel that this manuscript is now acceptable for publication, you may indicate that here to bypass the “Comments to the Author” section, enter your conflict of interest statement in the “Confidential to Editor” section, and submit your "Accept" recommendation.

Reviewer #1: All comments have been addressed

Reviewer #2: All comments have been addressed

2. Is the manuscript technically sound, and do the data support the conclusions?

Reviewer #1: Yes

Reviewer #2: Yes

3. Has the statistical analysis been performed appropriately and rigorously? 

Reviewer #1: Yes

Reviewer #2: Yes

4. Have the authors made all data underlying the findings in their manuscript fully available?

Reviewer #1: Yes

Reviewer #2: Yes

5. Is the manuscript presented in an intelligible fashion and written in standard English?

Reviewer #1: Yes

Reviewer #2: Yes

6. Review Comments to the Author

Reviewer #1: All comments are well addressed and it is much better than the previous one. However, I have few comments.

1 In discussion part "Finally, we did not record SvO2 which would have been valuable to compare oxygen consumption between groups". I think that this sentence should be change because we did not insert pulmonary arterial (PA) catheter in all patients and we will insert this PA catheter in case of pulmonary hypertension or in case that we would like to evaluate shunt effect in ARDS patients, May be you can write " we did not evaluate central venous oxygen saturation which was valuable to compare balance between oxygen consumption and oxygen delivery or you can say that we did not record central venous oxygen derived variables to compare oxygen consumption.

2 Please note Data was expressed in .... in Tables

Reviewer #2: Nice revision. Good luck! By adding the newest methods for analysing the microcirculation you make the follow-up for your study (and your patients ICU care) easier. "“A new tool called MicroTools software allowing an automated video analysis have been recently validated in a large international retrospective database [29]. This kind of automated analysis allowing resuscitation therapies targeted on microcirculatory parameters should be applied in further investigations.” Great sentence.

7. PLOS authors have the option to publish the peer review history of their article (what does this mean?). If published, this will include your full peer review and any attached files.

Reviewer #1: No

Reviewer #2: No

---

## [Author Response · Author response to Decision Letter 1]

19 Jan 2021

Response to reviewer 1

Reviewer #1: All comments are well addressed and it is much better than the previous one. 

Response: Thank you

However, I have few comments.

1 In discussion part "Finally, we did not record SvO2 which would have been valuable to compare oxygen consumption between groups". I think that this sentence should be change because we did not insert pulmonary arterial (PA) catheter in all patients and we will insert this PA catheter in case of pulmonary hypertension or in case that we would like to evaluate shunt effect in ARDS patients, May be you can write " we did not evaluate central venous oxygen saturation which was valuable to compare balance between oxygen consumption and oxygen delivery or you can say that we did not record central venous oxygen derived variables to compare oxygen consumption.

Response: Thank you. We changed the sentence in the manuscript. 

2 Please note Data was expressed in .... in Tables

Response: Thank you. We made changes in both Tables as follows: “Data were expressed as median [interquartile range] or as number (percentage)”

 

Response to reviewer 2

Reviewer #2: Nice revision. Good luck! By adding the newest methods for analysing the microcirculation you make the follow-up for your study (and your patients ICU care) easier. "“A new tool called MicroTools software allowing an automated video analysis have been recently validated in a large international retrospective database [29]. This kind of automated analysis allowing resuscitation therapies targeted on microcirculatory parameters should be applied in further investigations.” Great sentence.

Response: thank you.

---

## [Editor Report · Decision Letter 2]

25 Jan 2021

Microvascular flow alterations  in critically ill COVID-19 patients: a prospective study

PONE-D-20-32823R2

Dear Dr. abou arab,

We’re pleased to inform you that your manuscript has been judged scientifically suitable for publication and will be formally accepted for publication once it meets all outstanding technical requirements.

Kind regards,

Yu-Chang Yeh, M.D., Ph.D.

Academic Editor

PLOS ONE
---

## [Editor Report · Acceptance letter]

27 Jan 2021

PONE-D-20-32823R2 

Microvascular flow alterations in critically ill COVID-19 patients: a prospective study 

Dear Dr. Abou-Arab:

I'm pleased to inform you that your manuscript has been deemed suitable for publication in PLOS ONE. Congratulations! Your manuscript is now with our production department. 

Kind regards, 

on behalf of

Dr. Yu-Chang Yeh 

Academic Editor

PLOS ONE